# Darier Disease with Psoriasis

**DOI:** 10.3390/medicina58070902

**Published:** 2022-07-06

**Authors:** Seok-Young Kang, So-Yeon Lee, Jin-Seo Park, Jin-Cheol Kim, Bo-Young Chung, Chun-Wook Park, Hye-One Kim

**Affiliations:** Department of Dermatology, Kangnam Sacred Heart Hospital, Hallym University, Seoul 24252, Korea; tjdjrdud@naver.com (S.-Y.K.); minggijeook@gmail.com (S.-Y.L.); pahajs@gmail.com (J.-S.P.); aiekfne@naver.com (J.-C.K.); victoryby@naver.com (B.-Y.C.); dermap@hanmail.net (C.-W.P.)

**Keywords:** Darier disease, Darier-White disease, keratosis follicularis, dyskeratosis follicularis, psoriasis

## Abstract

Darier disease is an autosomal dominant disorder with dark crusty patches and is classified as hereditary acantholytic dermatosis. Keratotic papules and crust are often present on the scalp, forehead, chest, back, upper arms, elbows, groin, and behind the ears, predominantly in seborrheic areas. A 48-year-old male patient presented skin lesions with pruritus on the trunk and both upper and lower extremities. He first noticed the lesion 15 years before. On physical examination, there were multiple erythematous papules with crust on the trunk and red-brown colored keratotic plaque on both extremities. The suspected histopathological diagnosis was psoriasis vulgaris. The patient’s skin lesions and pruritus were significantly improved after the psoriasis treatment. While continuing psoriasis treatment, the patient showed sudden worsening of the skin lesions on the scalp, abdomen, and fingernails (V-shaped nicks) with pruritus. Punch biopsy was performed on the abdominal lesion again and the final diagnosis was Darier disease. The patient was then treated using alitretinoin while maintaining the use of guselkumab for psoriasis. There are only a few cases that we found in which patients with Darier disease also had psoriasis. We report this rare case of Darier disease with psoriasis and propose that an additional biopsy might be necessary for accurate diagnosis and proper treatment.

## 1. Introduction

Darier disease is a rare autosomal dominant inheritance disease, first reported by Darier and White in 1889. The prevalence of Darier disease is reported to range from 1:30,000–100,000, which is relatively rare [1]. Darier disease often develops in childhood and persists throughout adolescence with dark crusty patches that are greasy and malodorous. It is classified as hereditary acantholytic dermatosis [1]. Keratotic papules and crust are often present on the scalp, forehead, chest, back, upper arms, elbows, groin, and behind the ears, predominantly in seborrheic areas [1]. There are several diseases (such as seborrheic dermatitis, Hailey–Hailey disease, and acanthosis nigricans) that should be differentiated from Darier disease clinically and histopathologically [2,3]. Darier disease and psoriasis have common genetic risk factors involved in their pathophysiology, and more discussion might be needed. There are only a few cases that we found in which patients with Darier disease also had psoriasis [4].

## 2. Case Report

A 48-year-old male patient presented skin lesions with pruritus on the trunk and both upper and lower extremities. He first noticed the lesion 15 years before. The patient was treated for a long time at a local clinic for seborrheic dermatitis with little improvement. The patient’s family did not have any malignancies or other specific disease history. On physical and clinical examination, there were multiple erythematous papules with crust on the trunk. In addition, both upper and lower extremities exhibited red-brown colored keratotic plaque (Figure 1A). The patient gave us written consent to publish all photographic materials. Atopic dermatitis, nummular eczema, and psoriasis were clinically suspected, and a punch biopsy was performed on a skin lesion on the left upper arm. In the histopathologic examination, hyperkeratosis, acanthosis, and rete ridge elongation were present in the epidermis (H&E Scout view, Figure 1B). Psoriasiform hyperplasia, vasodilatation of the papillary dermis, and lymphocyte infiltration were observed in the dermis (H&E 100×, Figure 1C). In the epidermis at high magnification (H&E 400×, Figure 1D), Munro’s micro abscess with neutrophil dominance and parakeratosis was observed in the stratum corneum.

The suspected histopathological diagnosis was psoriasis vulgaris. The patient started treatment with cyclosporine (150 mg/day) and narrow-band ultraviolet B every week for three months. After that, the patient’s skin lesions and pruritus were significantly improved and the patient himself discontinued visiting as an outpatient for treatment. Four months after discontinuation of treatment, he visited as an outpatient due to aggravation of the prior skin lesions and resumed treatment with cyclosporine (100 mg/day). Despite continuous treatment, the hand area deteriorated (Figure 2A) and additional treatment with alitretinoin (30 mg/day) and excimer laser was performed. After 8 months of treatment, partial skin lesions improved, and after ixekizumab was administered to the patient for psoriasis, skin lesions of both upper and lower extremities and pruritus were significantly improved (Figure 2B). While continuing the ixekizumab treatment for 18 months, the patient showed sudden worsening of the skin lesions on the scalp, abdomen, and fingernails (V-shaped nicks) with pruritus (Figure 2C).

The treatment was changed to guselkumab, after which he improved for 4 months; however, the condition of the skin lesions waxed and waned repeatedly. A punch biopsy was performed on the abdominal lesion. Focal acantholysis was present in the suprabasal layer of the epidermal erosion area (Figure 3A). Dyskeratosis and corps ronds, pyknotic nuclei, and a clear perinuclear halo (Figure 3B) were observed in the epidermis. Corps grain compressed cells with elongated nuclei were seen in the stratum corneum and granular layer (Figure 3C).

The final diagnosis of the abdominal lesion was Darier disease, and the patient was then treated using alitretinoin (30 mg/day) while maintaining the use of guselkumab for psoriasis. The patient has been followed up and is now an outpatient with complete remission of psoriasis on the upper and lower extremities without recurrence. However, skin lesions on the scalp and abdomen with pruritus continued to wax and wane repeatedly. The patient recently presented partial remission with the addition of isotretinoin for the treatment of Darier disease and receives ongoing outpatient treatment.

## 3. Discussion

Darier disease is a rare autosomal dominant inherited disease and there are over 100 identified familial and sporadic mutations in the ATP2A2 gene and the entire mechanism of action is not clearly understood [1]. Darier disease shows a constellation of clinical findings involving the skin, nails (nail fragility, V-shaped nicks), and mucous membranes (multiple white papules similar to cobblestone lesions). Darier disease often develops in childhood and persists throughout adolescence with dark crusty patches that are greasy and malodorous. It is classified as hereditary acantholytic dermatosis. Some patients have exhibited evidence of the disease as early as four years old [2]. Histologically, Darier disease is characterized by dyskeratosis, such as corps ronds and grains, in addition to acantholysis leading to suprabasal cleavage. Particularly in the affected skin, internalization of adhesive molecules such as desmoglein I/II, desmocollin, and plakoglobin may be observed in epidermal keratinocytes that have undergone acantholysis with the cohesion of keratin fibers. These histologic features differentiated this from other diseases [3].

There is only one case reported in which a 60-year-old female patient with Darier disease also had psoriasis vulgaris [4]. She had dark brown, warty papular lesions in seborrheic distribution and large erythematous plaques with dry silver-colored scales on both lower extremities simultaneously. Several diseases should be differentiated from Darier disease clinically and histopathologically [3]. Darier disease is often undiagnosed or misdiagnosed as seborrheic dermatitis leading to undue stress for the patient and improper treatment. The 23-year-old patient in the case had been misdiagnosed and treated for seborrheic dermatitis and eczema for at least 2 years, similar to the patient in our case [5]. Darier disease is commonly misdiagnosed as eczema, dermatitis, seborrheic dermatitis, Hailey–Hailey disease, or acanthosis nigricans [5]. However, Darier disease is easily distinguished from other differential diagnoses in that it shows the nail and mucous membrane involvement as described above. Seborrheic dermatitis shows no nail involvement and histologically, it shows a complete absence of acantholytic dyskeratosis. The Hailey–Hailey disease usually does not show nail involvement and histologically, it is distinguished by the characteristic appearance of a dilapidated brick wall. In acanthosis nigricans, the lesion is usually more pigmented [6]. Acrokeratosis verruciformis of Hopf, a localized disorder of keratinization of distal extremities, is closely related to Darier disease and appears to be caused by mutations in the same gene. Histologically, the disease needs to be differentiated from benign familial pemphigus, Grover’s disease, Hailey–Hailey disease, and pemphigus vulgaris [5].

Treatments for Darier disease include moisturizers, urea, and lactic acid to relieve hyperkeratosis [1]. In addition, topical steroids are used as topical agents, and acitretin, isotretinoin, and alitretinoin may be considered as oral medications [2]. The patient, in this case, continued to use guselkumab as a treatment for psoriasis, added alitretinoin, and showed partial remission after the addition of isotretinoin as symptoms improved and worsened repeatedly.

Some studies on the association between Darier disease and psoriasis have been reported. Darier disease is known to be a problem linked to mutations in the gene ATP2A2, and there were some reports that psoriasis was also found to be associated with the gene ATP2A2 [7]. One large-scale gene expression study found that patients with psoriasis had a down-regulation of ATP2A2 compared with controls [7]. Darier disease and psoriasis have in common abnormalities in the calcium metabolism of cells, but it was confirmed that the TRPC1 expressed as a result was decreased in psoriasis and increased in Darier’s disease [8]. In both diseases, endoplasmic reticulum stress and the resulting unfolded protein reaction (UPR) are involved in pathogenesis [9,10]. Abnormal expression of involucrin, a protein related to keratinocyte differentiation, was similarly observed [11]. Cases of patients with both Darier disease and psoriasis have rarely been reported, and further studies are needed on the relationship between the two diseases.

## 4. Conclusions

Darier disease is apt to be confused with other dermatologic diseases and misdiagnosed, so skin biopsy should be performed initially. Thus, we report this rare case of Darier disease with psoriasis and propose that depending on the patient, an additional biopsy might be necessary for accurate diagnosis and proper treatment.

## Figures and Tables

**Figure 1 medicina-58-00902-f001:**
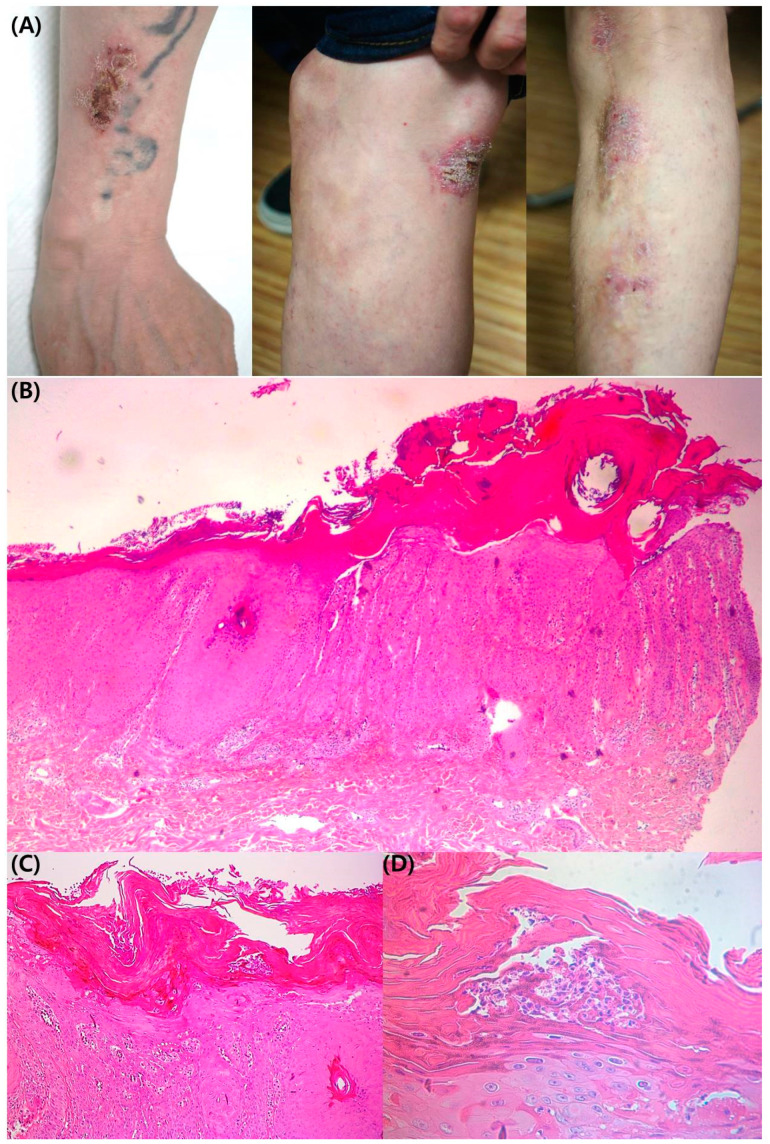
(**A**) Red-brown colored keratotic plaque with crust on the trunk, and both upper and both lower extremities. (**B**) Hyperkeratosis, acanthosis, and rete ridge elongation were present in the epidermis (H&E Scout view). (**C**) Psoriasiform hyperplasia, vasodilatation of the papillary dermis. (**D**) In high- magnification images of the epidermis, Munro’s microabscess with neutrophil dominance and parakeratosis appeared in the stratum corneum (H&E 400×).

**Figure 2 medicina-58-00902-f002:**
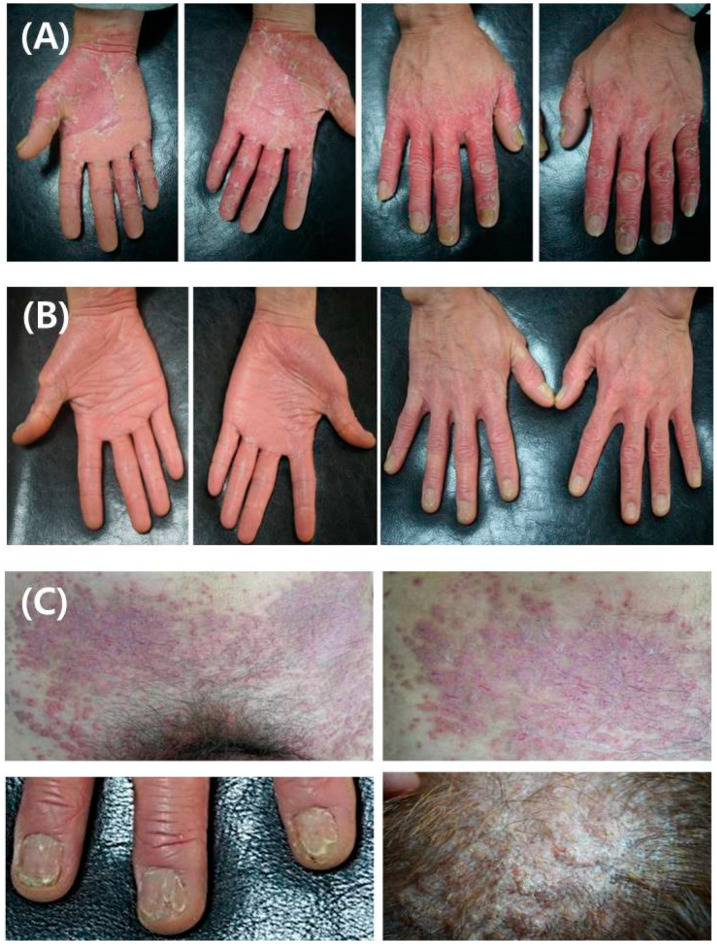
(**A**) Despite continuous treatment for psoriasis vulgaris, the skin lesions on the hands deteriorated. (**B**) After ixekizumab was administered to the patient for psoriasis, pruritus and skin lesions in both upper and lower extremities were significantly improved. (**C**) While continuing ixekizumab treatment for 18 months, the patient showed sudden worsening skin lesions on the scalp, abdomen, and finger nails (V-shaped nicks) with pruritus.

**Figure 3 medicina-58-00902-f003:**
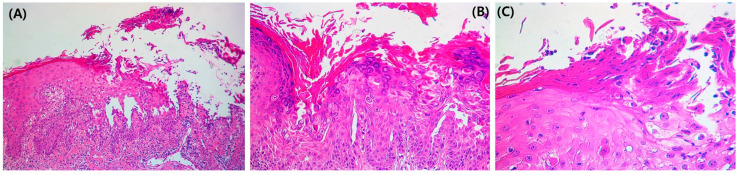
(**A**) Focal acantholysis was present in the suprabasal layer of the erosion area in the epidermis (H&E 100×). (**B**) Dyskeratosis and corps ronds, pyknotic nuclei, and a clear perinuclear halo (H&E 200×) were observed in the epidermis. (**C**) Corps grain compressed cells.

## Data Availability

Not applicable.

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
