# Peer review of "Darier Disease with Psoriasis"

_medicina, 2022, doi:10.3390/medicina58070902_

Round 1

Reviewer 1 Report

The article by Kang et al, describes a rare and interesting case of Darier disease with psoriasis.

1.     Could the authors briefly discuss why they excluded all the mentioned differential diagnosis?

2.     Could the authors briefly discuss the therapy of the patient?

3.     The abovementioned points are parts of your conclusions, so you have to previously briefly explain them in the manuscript. You also mention the value of additional biopsies which of course is the golden standard for diagnosis.

4.     In case the authors have word count limitations, Lines 108-122 can be rephrased and summarized, so that the previous questions are answered.

Author Response

Thanks for your review. The reason for excluding all the mentioned differential diagnoses was added to the discussion, and treatment was also added to the discussion section.

Thank you.

Reviewer 2 Report

The authors describe a case of a patients affected by a rare disease in association with a more frequent skin disease. The combination of both this pathologies in a patients has been rarely reported. Although psoriasis is a well-known disease despite a little knowledge about its etiology, Darier disease is a rare genetic pathology. Due to the possibility of a misdiagnosis, sometimes it could be hard to treat this kind of patients.

The paper is clear and supported by a good description and an sufficient number of photos and tissue exams.

This report could help other dermatologist in a quick diagnosis.

Author Response

Thank you for your review.